# Sedentary Duration and Systemic Health Burden: Nonlinear Associations with Muscle, Fat, and Vascular Phenotypes in a US Population-Based Study

**DOI:** 10.3390/healthcare13182309

**Published:** 2025-09-16

**Authors:** Chen Hu, Yang Song, Dong Sun, Zhenghui Lu, Hairong Chen, Xuanzhen Cen, Danica Janićijević, Zsolt Radak, Zixiang Gao, Julien Steven Baker, Yaodong Gu

**Affiliations:** 1Faculty of Sports Science, Ningbo University, Ningbo 315211, China; 2311040005@nbu.edu.cn (C.H.); sundong@nbu.edu.cn (D.S.); cenxuanzhen@outlook.com (X.C.); 2Department of Biomedical Engineering, Faculty of Engineering, The Hong Kong Polytechnic University, Hong Kong 999077, China; yangsong@polyu.edu.hk; 3Faculty of Engineering, University of Pannonia, 8200 Veszprem, Hungary; luzhenghui_nbu@foxmail.com; 4Doctoral School on Safety and Security Sciences, Óbuda University, 1034 Budapest, Hungary; chenhairong233@163.com; 5Department of Physical Education and Sport, Faculty of Sport Sciences, University of Granada, 18012 Granada, Spain; jan.danica@gmail.com; 6Research Institute of Sport Science, Hungarian University of Sport Science, 1525 Budapest, Hungary; radak.zsolt@tf.hu; 7Human Performance Laboratory, Faculty of Kinesiology, University of Calgary, Calgary, AB T2N 1N4, Canada; zixiang.gao@ucalgary.ca; 8Biomechanics Laboratory, Division of Sports and Exercise, University of the West of Scotland, Glasgow G72 0LH, UK; jsbaker@hkbu.edu.hk

**Keywords:** sedentary behavior, sarcopenia, body fat distribution, hemodynamics, cross-sectional studies

## Abstract

**Background:** Sedentary behavior (SB) is a growing public health concern associated with cardiometabolic risk; yet few studies have assessed integrated physiological responses across the muscle–fat–vascular system. **Methods:** This retrospective cross-sectional analysis used data from 13,637 participants (≥12 years) in the 2011–2018 National Health and Nutrition Examination Survey (NHANES). Sedentary duration (SD) was self-reported via a validated questionnaire. Outcomes included the sarcopenic index (SI), fat distribution index (FDI), and pulse pressure index (PPI). Associations were examined using multivariable linear regression and restricted cubic spline models, adjusting for sociodemographic and lifestyle factors. Subgroup analyses explored effect modification by body mass index (BMI), sex, race/ethnicity, education, and self-rated health. **Results:** Each additional hour/day of SD was associated with a lower SI (β = −0.004, 95% CI: −0.005 to −0.002), lower FDI (β = −0.009, 95% CI: −0.012 to −0.007), and higher PPI (β = 0.001, 95% CI: 0.000 to 0.002). The SD–SI association was nonlinear, with a threshold at 10.73 h/day: below this point, the SI declined sharply (β = −0.001, *p* < 0.001), while above it the slope plateaued or reversed. The FDI showed consistent adverse associations across the SD range, particularly in men and individuals with lower education. The PPI was significantly elevated with SD only among non-Hispanic Black participants. **Conclusions:** SD is differentially associated with muscle mass, fat distribution, and vascular function, with overlapping inflection points indicating a coordinated multisystem response to sedentary stress. These findings support targeting <10.7 h/day sedentary time as a potential intervention threshold.

## 1. Introduction

Sedentary behavior (SB) refers to any waking activity performed in a seated or reclining posture with very low energy expenditure, typically less than 1.5 metabolic equivalents, placing it at the lowest end of the physical activity continuum [1]. A growing body of epidemiological evidence has linked SB with a wide range of adverse health outcomes, including cardiovascular disease, metabolic dysfunction, cognitive decline, adverse pregnancy outcomes, and increased all-cause mortality. Previous studies have demonstrated that sedentary duration (SD) is negatively associated with various muscle performance parameters, such as handgrip strength and appendicular lean mass [2]. And it is positively correlated with obesity prevalence and the incidence of adverse cardiovascular events [3]. Interestingly, many studies have shown that physical activity benefits the musculoskeletal and metabolic systems; however, there is evidence indicating that physical activity levels do not significantly influence the association between sedentary behavior and adverse metabolic and cardiovascular outcomes [4,5]. Additionally, prolonged SB has been shown to significantly increase systolic blood pressure and mean arterial pressure [6]. These associations are particularly pronounced among adolescents, office workers, and older adults, highlighting the urgent need for systematic evaluation and mechanistic exploration of the multisystem health impacts of SB [7,8,9]. With the ongoing advancement of technology and evolving lifestyle patterns, SD has continued to rise globally, prompting the World Health Organization to classify it as a major public health threat [10,11,12,13].

However, the relationship between sedentary behavior and the cardiovascular and metabolic systems is complex and closely interconnected. Studies have shown that sedentary behavior is associated with adverse outcomes such as muscle atrophy, fat accumulation, and blood pressure dysregulation [9,14,15,16]. The accumulation of these adverse events further impairs metabolic homeostasis and vascular function, manifesting as reduced tissue function, insulin resistance, and arterial stiffness [17]. Comprehensively elucidating the relationships between sedentary behavior and the three major systems—muscle, fat, and blood flow—remains a challenging task.

Previous studies have consistently demonstrated that skeletal muscle mass is also a key predictor of quality of life, disability risk, and all-cause mortality [18,19,20,21]. Similarly, fat distribution patterns and hemodynamic characteristics are strongly associated with disease risk and overall health status [22,23,24,25,26,27]. Unfortunately, the existing literature has largely examined these systems in isolation, lacking an integrated framework that accounts for their interconnected responses to SB. This fragmented research perspective limits our understanding of the systemic effects of SB and hampers the development of evidence-based and multidimensional intervention strategies [28].

Moreover, it is important to recognize that the relationship between SB and alterations in muscle, fat, and blood flow may be nonlinear. Accumulated SD may exhibit threshold effects or plateau phases, wherein risk escalates more sharply within specific exposure ranges. Failure to account for such characteristics may lead to inaccurate risk estimation, while overly coarse time categorizations may obscure critical inflection points [29]. Identifying potential nonlinear associations between SD and multisystem biomarkers is therefore essential for the refinement of public health guidelines [30].

Drawing on data from the National Health and Nutrition Examination Survey (NHANES), this study aims to examine the potential relationship between SB and three key physiological systems—skeletal muscle, adipose tissue, and hemodynamic function (Figure 1). We seek to construct models that characterize these associations and to explore possible nonlinear patterns. By doing so, we aim to provide evidence-based insights to inform SB intervention strategies and to advance the development of integrative biomarker frameworks for chronic disease prevention. We hypothesize that shorter daily SD is associated with higher skeletal muscle mass, healthier fat distribution, and more favorable hemodynamic profiles.

## 2. Materials and Methods


**Study Population and Inclusion Criteria**


This study analyzed retrospective data from four consecutive cycles (2011–2018) of the NHANES. The NHANES is a major program designed to assess the health and nutritional status of the non-institutionalized U.S. civilian population, conducted by the National Center for Health Statistics (NCHS). The NHANES data were accessed via the official website of the Centers for Disease Control and Prevention (CDC) (https://www.cdc.gov/nchs/nhanes/) (accessed on 26 March 2025). All datasets are publicly available and de-identified, and ethical approval was obtained by the NCHS. Therefore, no additional institutional review board approval was required for this secondary analysis.

The NHANES data for this study were collected by trained professionals at the CDC during home interviews and mobile examination center (MEC) visits. For the present analysis, two independent researchers (C.H. and Y.S.) reviewed the datasets, applied the predefined inclusion and exclusion criteria to screen participants, and cross-checked the coding of all variables to ensure accuracy and reproducibility. The data collection and cleaning workflow are illustrated in Figure 2.

As is shown in Figure 2, a total of 39,156 participants were initially included in this study. After sequentially excluding individuals without body measurement data (*n* = 4728), those without dual-energy X-ray absorptiometry (DXA) data (*n* = 15,657), those with incomplete anthropometric data (*n* = 2874), and individuals with abnormal anthropometric or questionnaire data (*n* = 2366), a final sample of 15,321 participants remained for analysis. After applying the appropriate sample weights, this analytic population represents approximately 135,218,341 person-visits.


**Exposure and Outcome Variables**


The selection of exposure and outcome variables was primarily based on previous studies and finalized after thorough discussion and consensus among all authors [31,32,33,34,35].

The primary exposure variable was daily SD, assessed via the Computer-Assisted Personal Interview (CAPI) system by trained interviewers during in-home interviews. The participants were asked: “The following question is about the time you spend sitting at school, at home, getting to and from places, or with friends, including time spent sitting at a desk, traveling in a car or bus, reading, playing cards, watching television, or using a computer. Do not include time spent sleeping. On a typical day, how much time do you usually spend sitting?”

To more clearly elucidate the associations between sedentary behavior and muscle mass, fat distribution, and hemodynamic characteristics, this study prioritized the use of continuous variables that could be normalized for analysis. Based on previous literature, data availability, and the quality of epidemiological evidence, three primary outcome variables were selected to represent the musculoskeletal, adipose, and hemodynamic systems: the sarcopenic index (SI), the fat distribution index (FDI), and the pulse pressure index (PPI), respectively [31,32,33,34,35]. As a simple, cost-effective, and objective indicator, the SI shows a significant correlation with actual muscle mass and holds substantial clinical relevance [36]. The SI was calculated as the total appendicular skeletal muscle mass (kg) divided by the BMI (kg/m^2^). The FDI is more effective than body weight or body fat percentage alone in predicting the risk of metabolic syndrome, including cardiovascular disease and diabetes, and has broad applicability [37]. The FDI was defined as the ratio of trunk fat mass to appendicular fat mass. The PPI not only reflects arterial stiffness and vascular aging but also serves as an important reference indicator for risk assessment and early intervention of chronic diseases such as cardiovascular disease and impaired glucose metabolism [38]. The PPI was computed as pulse pressure divided by systolic blood pressure [35,39,40,41,42,43,44]. Muscle and fat mass in the trunk and limbs were measured by CDC professionals using DXA, which is widely accepted for body composition assessment due to its speed, ease of use, and low radiation exposure [45,46]. Blood pressure was measured in the MEC after participants had rested in a seated position for at least five minutes and after determining their maximum inflation level (MIL) [47].


**Covariates**


Covariates were primarily selected based on prior literature and finalized through thorough discussion and consensus among all authors, encompassing both categorical and continuous variables [48,49,50,51]. Categorical variables included sex, race/ethnicity, educational attainment, marital status, self-rated health, and perceived body weight status. Continuous variables included age, alcohol consumption, tobacco use, sleep duration, body mass index (BMI), waist circumference, body surface area, systolic blood pressure, and diastolic blood pressure [48,49,50,51].


**Statistical Analysis**


Statistical analyses consisted of three interrelated components designed to comprehensively assess the relationships between SD and indicators of muscle loss, fat redistribution, and hemodynamic function. The participants were stratified into tertiles based on their reported daily SD, and baseline characteristics were summarized accordingly. Continuous variables were presented as mean ± standard deviation, and categorical variables as percentages. Group differences were evaluated using the χ^2^ test for categorical variables, one-way ANOVA for normally distributed continuous variables, and the Kruskal–Wallis H test for skewed continuous variables.

Associations between SD and the three system indices were examined using multivariable linear regression models. Following the STROBE guidelines, three models were constructed: Model 1 (unadjusted); Model 2 (adjusted for age, sex, race/ethnicity, education level, and marital status); and Model 3 (fully adjusted for all identified covariates) [52,53]. This stepwise modeling approach facilitated the evaluation of potential confounding effects. Stratified linear regression models were also applied to explore possible effect modifications, and likelihood ratio tests were used to assess interactions among subgroup variables [54,55].

To account for potentially complex relationships, restricted cubic spline (RCS) models were used to explore and confirm nonlinear associations between SD and the three system indices, with threshold effect analyses employed to identify potential inflection points [56,57].

Missing data were handled using multiple imputation by chained equations, which is a robust multilevel method specifically designed for survey data [58,59]. Ten imputed datasets were generated using Gibbs sampling after 500 iterations and 100 burn-in updates to ensure independence across imputations [60]. All statistical analyses were performed using R software (version 4.4.2). The RCS models were fitted using the rms package in R, with knots placed at the 10th, 50th, and 90th percentiles [61]. A two-sided *p*-value < 0.05 was considered statistically significant.

## 3. Results

### 3.1. Baseline Characteristics of the Study Participants

The detailed baseline characteristics of the participants are presented in Appendix A. The participants were regrouped into tertiles based on daily SD, from shortest to longest. The results indicated no significant differences among groups in terms of gender, sleep duration, self-rated health status, trunk fat percentage, leg fat mass, or pulse pressure. However, significant differences were observed across groups in age, race, education level, marital status, daily alcohol intake, daily tobacco use, self-perceived body weight, systolic blood pressure, diastolic blood pressure, body surface area, total bone mass, total bone density, overall fat percentage, appendicular skeletal muscle mass, trunk fat mass, arm fat mass, BMI, waist circumference, SI, FDI, and PPI.

### 3.2. Association Between SD and Sarcopenia, Fat Distribution, and Hemodynamics

Table 1 presents the results of multivariable regression analyses exploring the associations between SD and the SI, FDI, and PPI. In the unadjusted model, SD was strongly negatively associated with both the SI and FDI (*p* < 0.001), and significantly positively associated with the PPI (*p* < 0.05). In the fully adjusted model, accounting for potential confounding factors, SD remained significantly negatively associated with the SI (β = −0.001, 95% CI: −0.001 to 0.000, *p* < 0.05) and showed a strong negative association with the FDI (β = −0.003, 95% CI: −0.005 to −0.002, *p* < 0.001). However, no significant association was found between SD and the PPI after full adjustment.

### 3.3. Stratified Analysis of the Associations

Stratified analyses were conducted to explore the associations between SD and sarcopenia, fat distribution, and hemodynamic indicators across various subgroups defined by gender, race, education level, marital status, self-rated health, self-perceived body weight, BMI category, and hypertension status (Figure 3). The results of the subgroup post hoc tests are presented in Table 2. The association between SD and the SI was significantly modified by self-perceived body weight (*p* = 0.003, the More and Don’t know groups demonstrated post hoc statistical power above 80%, while the Less and Same groups showed power below 80%) and BMI category (*p* = 0.000, the Underweight and Normal-weight groups both exhibited post hoc statistical power greater than 80%, whereas the Overweight and Obese groups had power values below 80%). The association between SD and the FDI was significantly modified by gender (*p* = 0.005, the Male group demonstrated post hoc statistical power above 80%, while the Female group showed power below 80%), race (*p* = 0.001, the Mexican American, Other Hispanic, and Non-Hispanic White groups all had post hoc statistical power greater than 80%, whereas the Non-Hispanic Black, Non-Hispanic Asian, and Other Race—including multi-racial—groups had power below 80%), education level (*p* = 0.039, the Less than 9th grade, 9–11th grade, and Some college or associate degree groups had post hoc statistical power above 80%, while the High school graduate and College graduate or above groups had power below 80%), and self-rated health (*p* = 0.005, the Good and Fair groups showed post hoc statistical power above 80%, whereas the Excellent, Very good, and Poor groups showed power below 80%). The association between SD and PPI was significantly modified by race (*p* = 0.029; all categories demonstrated post hoc statistical power below 80%).

### 3.4. Nonlinear Relationships Between SD and Muscle–Fat–Flow Indicators

To investigate the hypothesis of nonlinear relationships between SD and changes in muscle mass, fat distribution, and hemodynamics, restricted cubic spline regression was used. Models were constructed with three levels: unadjusted; adjusted for age, sex, race, education, and marital status; and fully adjusted for additional factors, including alcohol consumption, sleep duration, smoking status, self-rated health, self-perceived body weight, BMI, waist circumference, body surface area, and systolic and diastolic blood pressure. As shown in Figure 4, SD was significantly associated with the SI, FDI, and PPI. All models, except for the fully adjusted SD–PPI model, showed strong associations (*p* < 0.001).

Furthermore, in the unadjusted models, SD exhibited highly significant nonlinear relationships with all three outcome indices (*p* for nonlinear < 0.001). After adjusting for age, sex, race, education, and marital status, the nonlinear associations persisted for the SI and FDI (*p* for nonlinear < 0.05). In the fully adjusted models, a significant nonlinear relationship remained only between SD and the SI (*p* for nonlinear < 0.05).

Threshold effect analysis was used to determine the inflection points for the nonlinear models, with specific details provided in Table 3. In the unadjusted models, if daily SD was less than 9.467 h, it was negatively associated with all three indices. If SD exceeded 9.467 h, it became positively associated. After adjusting for age, sex, race, education, and marital status, the threshold was 10.517 h. In the fully adjusted model, the threshold was 10.733 h. Below this threshold, SD was negatively associated with the SI; above it, SD was positively associated with the SI and FDI.

### 3.5. Stratified RCS Analysis in Positive Subgroups

To further examine whether nonlinear dose-response relationships existed within specific positive subgroups, RCS models were stratified by key modifiers (Figure 5). The correlation between SD and the SI was significantly and nonlinearly modified by self-perceived body weight (*p* for overall < 0.001, *p* for nonlinear < 0.05). In other positive subgroups related to muscle, fat, and flow indicators, significant associations were observed across different subgroup levels, but these relationships were linear (*p* for overall < 0.001, *p* for nonlinear > 0.05).

## 4. Discussion

Using retrospective data from the NHANES, this study systematically examined the associations of sedentary behavior with the degree of muscle loss, patterns of fat distribution, and hemodynamic characteristics, and provided a preliminary interpretation of the potential underlying mechanisms. Overall, the linear model results revealed that SD was negatively associated with the muscle loss index and FDI, and positively associated with the PPI. These findings suggest that longer SD may be linked to greater skeletal muscle degradation, increased central obesity, and potentially reduced vascular compliance. The robust associations with muscle loss and fat redistribution may be related to reduced myokine secretion and the accumulation of metabolic disturbances during sedentary periods [62,63]. Although the association with pulse pressure weakened after multivariable adjustment, this trend aligns with the typical attenuation of effect sizes due to confounding in large-scale studies [64].

Further subgroup analyses indicated that various demographic and sociological factors played important roles within the correlation models. The association between SD and the muscle loss index was jointly influenced by the BMI and self-perceived body weight, suggesting that perceptual bias may alter health impacts by affecting exercise motivation [65,66,67]. Associations with the FDI were moderated by race, gender, education level, and self-rated health, reflecting complex differences in the nature, motivation, and accompanying behaviors (e.g., diet and physical activity) of SD across populations [68,69,70]. Interestingly, SD was positively associated with fat redistribution in men but negatively in women—though this gender difference disappeared after broader covariate adjustment, emphasizing the role of social and behavioral context in mediating SB’s health effects. Moreover, the association between SD and the PPI was mainly observed in non-Hispanic Black individuals, suggesting that genetic background and vascular biology may play modulatory roles in different racial groups [71,72].

In this study, the presence of a threshold indicates that the strength of the association between SD and a given outcome reaches a maximum or minimum at that point, with the direction of the association reversing before and after the threshold. We observed a nonlinear association between SD and the muscle loss index. Specifically, the risk of muscle loss increased with longer SD, but this trend reversed once SD exceeded a threshold of 10.73 h/day. This phenomenon may be related to the activation of certain compensatory mechanisms in the body, such as engaging alternative metabolic pathways to slow further muscle loss. At the same time, once sedentary time exceeds this threshold, other factors—such as diet, genetic predisposition, and exercise habits—may become predominant, thereby leading to an apparent reversal of the trend. This nonlinear pattern remained stable across different adjustment models, suggesting that SB may be linked to deeper mechanisms, such as molecular-level changes in muscle gene expression, deterioration of muscle fiber function, and disruption of synergistic activation [73,74]. In contrast, the nonlinear associations of SD with the FDI and PPI weakened after multiple adjustments and ultimately exhibited stable linear trends. This divergence may reflect differences in biological response windows, threshold sensitivities, and adaptive mechanisms across systems under sedentary stress.

Moreover, after adjusting for demographic variables, we found a high degree of consistency in the threshold breakpoints for muscle, fat, and vascular indices, suggesting the possible existence of an underlying axis that collectively governs the associations between SD and the muscle–fat–vascular systems and their patterns of change. This mechanism may involve metabolic crosstalk among muscle, fat, and vasculature, chronic low-grade inflammation, and the co-regulation of shear stress–sensing pathways. These findings support the hypothesis that sedentary behavior constitutes a systemic health threat and underscore the need to establish population-specific sedentary time thresholds for early health risk screening. They also deepen our understanding of its health impacts and provide biological justification for future multisystem–targeted interventions.

This study leveraged the representative, large-scale NHANES data with integrated analysis strategies, including hierarchical subgrouping and nonlinear modeling. To some extent, this avoided potential biases such as the constituent year effect, which strengthen its robustness and generalizability [75]. However, several limitations remain. First, as this study employed a retrospective cross-sectional design, it can only assess the associations of SD with the SI, FDI, and PPI under different conditions. This design inherently precludes any causal inference and does not allow deductions regarding pathophysiological trajectories. Second, sedentary behavior was recorded through self-reported measures, which not only introduces recall bias but may also be influenced by social desirability bias. Such limitations could adversely affect the determination of inflection points in nonlinear relationships. Considering that device-based measurements are more accurate and not restricted by data collection periods, future research should, wherever possible, incorporate objective measurement tools [76]. In addition, this study involved multiple subgroups and stratified analyses; however, the statistical power in some subgroups was relatively insufficient due to the limited sample sizes. Constrained by the methodological nature of retrospective cross-sectional research, it is difficult to supplement the sample size further. Finally, although combining multiple consecutive cycles can substantially increase the analyzable sample size and thereby enhance statistical power, subtle differences in methodology, questionnaire design, or laboratory measurements across cycles may still exist, which could limit the absolute accuracy of the study findings. Therefore, the interpretation of certain subgroup analyses should be discussed in conjunction with confidence intervals. Moreover, despite extensive covariate adjustment, residual confounding cannot be entirely ruled out. Future research should focus on more longitudinal studies to provide stronger evidence for developing more effective sedentary behavior intervention strategies [77].

## 5. Conclusions

This population-based analysis identified nonlinear associations between SD and muscle, fat, and vascular function. Specifically, SD below 10.73 h/day was inversely associated with the SI, whereas above this threshold, the association reversed. SD was also inversely related to the FDI, while no significant association was found with the PPI. These findings partially support our initial hypothesis that sedentary behavior influences muscle mass and fat distribution, although the anticipated relationship with hemodynamic function was not observed. Further longitudinal and mechanistic studies are needed to confirm these associations and elucidate causal pathways.

## Figures and Tables

**Figure 1 healthcare-13-02309-f001:**
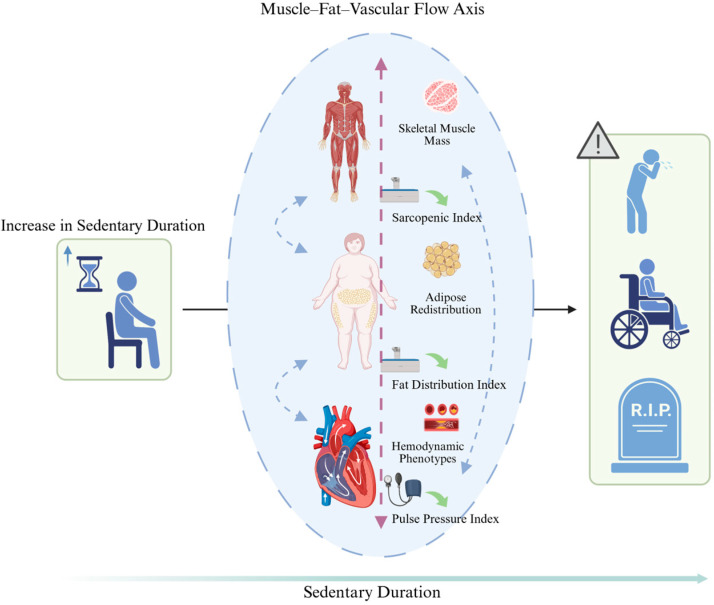
Potential Impacts of sedentary behavior on the muscle–fat–vascular flow axis.

**Figure 2 healthcare-13-02309-f002:**
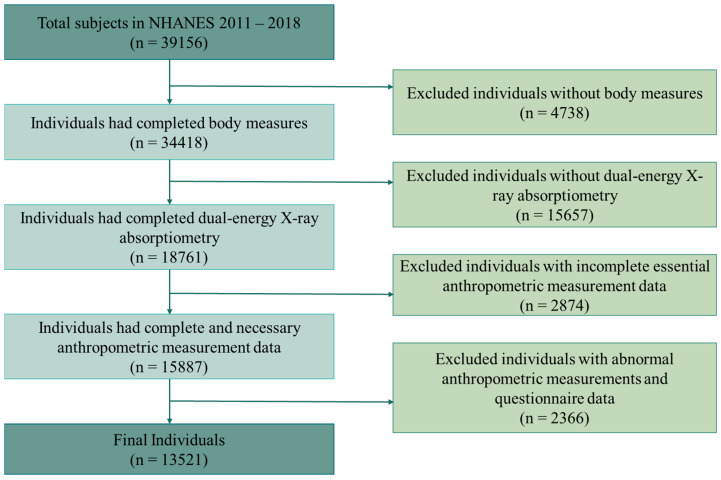
Flowchart of participant selection.

**Figure 3 healthcare-13-02309-f003:**
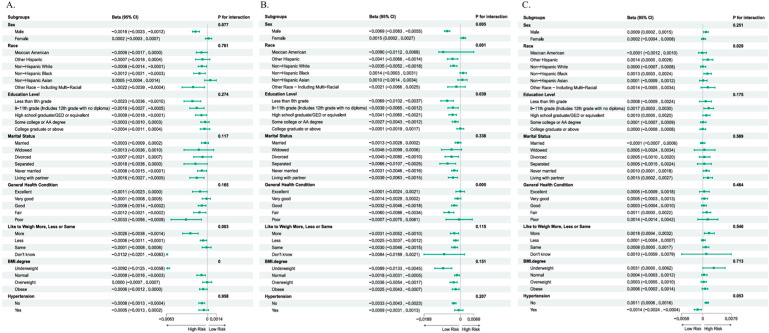
Stratified analysis of the relationship between SD and SI, FDI, and PPI. (**A**) represents the association between SD and SI; (**B**) represents the association between SD and FDI; (**C**) represents the association between SD and PPI.

**Figure 4 healthcare-13-02309-f004:**
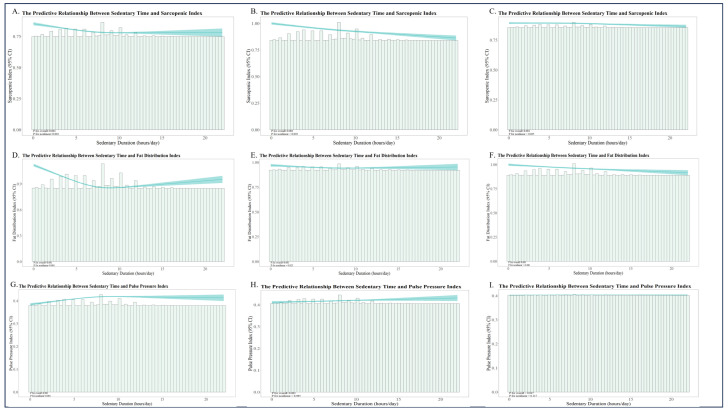
Restricted cubic spline analysis of SD and muscle–fat–hemodynamics. The models were adjusted as follows: unadjusted (**A**,**D**,**G**); adjusted for age, sex, race, education, and marital status (**B**,**E**,**H**); fully adjusted for age, sex, race, education, marital status, alcohol consumption, sleep duration, smoking status, self-rated health assessment, self-perceived weight status, BMI, waist circumference, body surface area, systolic blood pressure, and diastolic blood pressure (**C**,**F**,**I**).

**Figure 5 healthcare-13-02309-f005:**
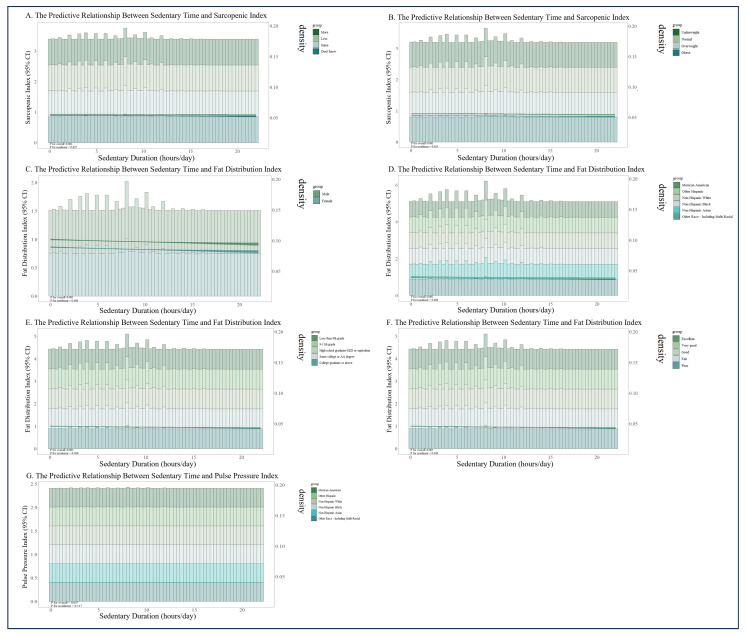
RCS curves of key indicators and their corresponding positive subgroups.

**Table 1 healthcare-13-02309-t001:** Relationship between SD and sarcopenia, fat distribution, and hemodynamics.

	Model 1			Model 2			Model 3		
	Beta	95%CI	*p*-Value	Beta	95%CI	*p*-Value	Beta	95%CI	*p*-Value
**SI**	−0.004	−0.005, −0.002	**<0.001**	−0.006	−0.007, −0.005	**<0.001**	−0.001	−0.001, 0.000	**0.004**
**FDI**	−0.009	−0.012, −0.007	**<0.001**	0.000	−0.002, 0.002	0.800	−0.003	−0.005, −0.002	**<0.001**
**PPI**	0.001	0.000, 0.002	**0.014**	0.000	0.000, 0.001	0.200	0.000	0.000, 0.000	0.200

Model 1: Unadjusted; Model 2: Adjusted for age, sex, race/ethnicity, educational attainment, and marital status; Model 3: Fully adjusted for all selected covariates, incorporating age, sex, race, education, marital status, alcohol consumption, sleep duration, smoking status, self-rated health assessment, self-perceived body weight, BMI, waist circumference, body surface area, systolic blood pressure, and diastolic blood pressure. SI, Sarcopenic Index. FDI, Fat Distribution Index. PPI, Pulse Pressure Index. Bolded numbers indicate *p* < 0.05.

**Table 2 healthcare-13-02309-t002:** Subgroup post hoc power analysis.

Different Subgroups	Category	n	β	95% CI	SE	t-Value	Power
**SI—Like to weigh more, less, or same**	More	1884	−0.0026	−0.0038~−0.0014	0.000612	−4.25	**98.9%**
	Less	7217	−0.0006	−0.0011~−0.0001	0.000255	−2.35	65.2%
	Same	4189	−0.0001	−0.0008~0.0006	0.000357	−0.28	5.9%
	Don’t know	231	−0.0132	−0.0201~−0.0063	0.003520	−3.75	**96.2%**
**SI—BMI degree**	Underweight	1132	−0.0092	−0.0125~−0.0058	0.00138	−6.67	**>99%**
	Normal weight	4896	−0.0009	−0.0016~−0.0003	0.00033	−2.73	**89%**
	Overweight	3648	0.0000	−0.0007~0.0007	0.00036	0	5%
	Obese	3846	−0.0006	−0.0012~0.0000	0.00031	−1.94	62%
**FDI—Sex**	Male	6801	−0.0069	−0.0083~−0.0055	0.00092	−7.50	**>99%**
	Female	6720	0.0015	0.0002~0.0027	0.00077	1.95	62%
**FDI—Race**	Mexican American	2330	−0.0090	−0.0112~−0.0069	0.00058	−15.52	**>99%**
	Other Hispanic	1434	−0.0041	−0.0068~−0.0014	0.00119	−3.44	**99%**
	Non-Hispanic White	4357	−0.0035	−0.0052~−0.0018	0.00069	−5.07	**>99%**
	Non-Hispanic Black	2838	0.0014	−0.0003~0.0031	0.00089	1.57	42%
	Non-Hispanic Asian	1869	0.0010	−0.0014~0.0034	0.00122	0.82	18%
	Other Race—including multi-racial	693	−0.0021	−0.0066~0.0025	0.00230	−0.91	20%
**FDI—Education level**	Less than 9th grade	1070	−0.0069	−0.0102~−0.0037	0.00133	−5.19	**>99%**
	9–11th grade (including 12th grade with no diploma)	1701	−0.0039	−0.0065~−0.0012	0.00143	−2.73	**89%**
	High school graduate/GEO or equivalent	2860	−0.0041	−0.0003~0.0031	0.00164	−0.25	6%
	Some college or AA degree	4235	−0.0027	−0.0043~−0.0012	0.00056	−4.82	**>99%**
	College graduate or above	3655	−0.0001	−0.0019~0.0017	0.00097	−0.10	5%
**FDI—General health condition**	Excellent	1604	−0.0001	−0.0024~0.0021	0.00118	−0.08	5%
	Very good	4152	−0.0014	−0.0029~0.0002	0.00088	−1.59	42%
	Good	5455	−0.0032	−0.0046~0.0018	0.00072	−4.44	**>99%**
	Fair	1951	−0.0060	−0.0086~−0.0034	0.00133	−4.50	**>99%**
	Poor	359	−0.0007	−0.0075~0.0061	0.00350	−0.20	6%
**PPI—Race**	Mexican American	2330	−0.0001	−0.0012~0.0010	0.00056	−0.18	6%
	Other Hispanic	1434	0.0014	0.0000~0.0028	0.00071	1.97	63%
	Non-Hispanic White	4357	0.0000	−0.0007~0.0008	0.00038	0.00	5%
	Non-Hispanic Black	2838	0.0013	0.0003~0.0024	0.00055	2.36	76%
	Non-Hispanic Asian	1869	0.0001	−0.0009~0.0012	0.00054	0.19	6%
	Other Race—including multi-racial	693	0.0014	−0.0005~0.0034	0.00101	1.39	33%

SI, Sarcopenic Index. FDI, Fat Distribution Index. PPI, Pulse Pressure Index. BMI, Body Mass Index. Bolded numbers indicate a statistical power greater than 80%.

**Table 3 healthcare-13-02309-t003:** Threshold effect analysis of nonlinear models.

Model	Effect Size	95%CI	*p* for Likelihood Ratio Test
**SD-SI—Model 1**			**<0.001**
Standard Linear Regression	−0.005	−0.006, −0.004	
Two-piecewise Linear Regression			
Inflection Point of ST			
<9.467	−0.01	−0.011, −0.008	
>9.467	0.01	0.006, 0.013	
**SD-SI—Model 2**			**<0.001**
Standard Linear Regression	−0.007	−0.008, −0.006	
Two-piecewise Linear Regression			
Inflection Point of ST			
<10.517	−0.01	−0.011, −0.009	
>10.517	0.011	0.008, 0.014	
**SD-SI—Model 3**			**<0.001**
Standard Linear Regression	−0.001	−0.001, 0	
Two-piecewise Linear Regression			
Inflection Point of ST			
<10.733	−0.001	−0.002, −0.001	
>10.733	0.002	0, 0.004	
**SD-FDI—Model 1**			**<0.001**
Standard Linear Regression	−0.005	−0.006, −0.004	
Two-piecewise Linear Regression			
Inflection Point of ST			
<9.467	−0.01	−0.011, −0.008	
>9.467	0.01	0.006, 0.013	
**SD-FDI—Model 2**			**<0.001**
Standard Linear Regression	−0.007	−0.008, −0.006	
Two-piecewise Linear Regression			
Inflection Point of ST			
<10.517	−0.01	−0.011, −0.009	
>10.517	0.011	0.008, 0.014	
**SD-PPI—Model 1**			**<0.001**
Standard Linear Regression	−0.005	−0.006, −0.004	
Two-piecewise Linear Regression			
Inflection Point of ST			
<9.467	−0.01	−0.011, −0.008	
>9.467	0.01	0.006, 0.013	

Model 1: Unadjusted; Model 2: Adjusted for age, sex, race/ethnicity, educational attainment, and marital status; Model 3: Fully adjusted for all selected covariates, incorporating age, sex, race, education, marital status, alcohol consumption, sleep duration, smoking status, self-rated health assessment, self-perceived body weight, BMI, waist circumference, body surface area, systolic blood pressure, and diastolic blood pressure. SI, Sarcopenic Index. FDI, Fat Distribution Index. PPI, Pulse Pressure Index. Bolded numbers indicate *p* < 0.05.

## Data Availability

The datasets generated during the current study are available on the NHANES website (https://www.cdc.gov/nchs/nhanes/) (accessed on 26 March 2025).

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
