# Peer review of "Sedentary Duration and Systemic Health Burden: Nonlinear Associations with Muscle, Fat, and Vascular Phenotypes in a US Population-Based Study"

_healthcare, 2025, doi:10.3390/healthcare13182309_

Round 1

Reviewer 1 Report

Comments and Suggestions for Authors

This study investigated the nonlinear associations of sedentary duration with muscle, fat, and vascular phenotypes in U.S. adults. The paper is well-written and based on sound methodology. Please consider the following suggestions for improvement:

  1. Introduction: The introduction is well-organized and effectively conceptualizes the study topic within an appropriate framework. However, the manuscript could benefit from a more comprehensive review of the existing literature on how SB was associated with sarcopenia, fat distribution, and hemodynamic outcomes (e.g., blood pressure, pulse pressure).
  2. Lines 136–141 : The authors included a broad range of covariates, which strengthens the model. However, the inclusion of variables that are closely linked to metabolic profiles, such as BMI, WC, SBP, and DBP, may be problematic, as they could obscure the associations between sedentary behavior and the primary outcomes. A more detailed rationale is needed to justify the inclusion of these variables.
  3. Lines 160–163: Since RCS models are central to the analysis, additional methodological details should be provided. Specifically, the number and placement of knots used, as well as the R package employed for the RCS analysis.
  4. The resolution of the figures is too low, and the images themselves are too small, making it difficult for readers to interpret the findings.
  5. Tables: abbreviations such as SI, FDR, and PPI should be defined in the table footnotes.
  6. In Figure 5, the curve appears too small to visually confirm its nonlinear pattern.
  7. The reversal in the association between SD and the muscle loss beyond the threshold warrants further elaboration. Please consider offering a hypothetical mechanism or citing relevant literature to help explain this pattern.
  8. SB was self-reported, which raises concerns not only about recall bias but also about social desirability bias. The limitations associated with self-reported sedentary time should be acknowledged, along with a note on the consideration of using objective measures in future studies.

Author Response

Dear Reviewer, thank you very much for your patience and suggestions concerning our manuscript entitled “Sedentary Duration and Systemic Health Burden: Nonlinear Associations with Muscle, Fat, and Vascular Phenotypes in a US Population-Based Study”. Those comments are all valuable and very helpful for revising and improving the quality of our paper, as well as the important guiding significance to our research. We have studied comments carefully and made point-to-point corrections. The revised portion is highlighted in red in the revised manuscript, and please find attached files for a point-by-point response to your comments and concerns.

Reviewer 2 Report

Comments and Suggestions for Authors

Summary of Strengths and Weaknesses

This population-based study presents an ambitious attempt to examine the multisystem effects of sedentary behavior using NHANES data from 2011-2018. The research addresses an important public health concern by investigating nonlinear dose-response relationships between sedentary duration and integrated physiological outcomes across muscle, adipose, and vascular systems.

Strengths:

The study's principal strength lies in its comprehensive approach to understanding sedentary behavior's systemic effects. The authors commendably constructed a theoretical framework linking sedentary duration to a "multisystem phenotype" using three composite indices. The use of restricted cubic spline modeling to identify nonlinear relationships and threshold effects represents sophisticated statistical methodology. The large, nationally representative sample (n=15,321) provides substantial statistical power, while the stratified analyses across demographic subgroups offer valuable insights into effect modification. The multiple imputation approach for handling missing data demonstrates methodological rigor.

Weaknesses:

However, several limitations substantially constrain the study's interpretability and impact. The cross-sectional design fundamentally limits causal inference, particularly problematic given the authors' discussion of "chronic stress responses" and "pathophysiological trajectories." The reliance on self-reported sedentary duration introduces significant measurement error and recall bias, especially concerning given the precision implied by the 10.73-hour threshold. The composite indices (SI, FDI, PPI), while conceptually interesting, lack established clinical validity and interpretability. The absence of sample size calculations raises questions about statistical power for subgroup analyses, particularly given the multiple testing burden inherent in stratified restricted cubic spline analyses.

Statistical Methodology Assessment

The statistical approach demonstrates both sophistication and concerning gaps. The restricted cubic spline methodology is appropriate for detecting nonlinear relationships, and the three-model hierarchical adjustment strategy follows best practices. However, the absence of formal sample size calculations represents a significant methodological oversight. Given the complexity of the stratified analyses and multiple testing across three outcomes and numerous subgroups, power calculations would be essential to interpret negative findings and establish the reliability of identified threshold effects. The multiple imputation approach using chained equations is well-executed, though the choice of ten imputed datasets could be better justified.

Enhanced Scientific Framework Through Strategic References

To strengthen this manuscript's scientific foundation and clinical relevance, several key references should be integrated strategically throughout the text. When discussing the physiological mechanisms underlying sedentary behavior's effects on muscle mass, the authors should incorporate Padulo et al.'s work on muscle activation patterns during different activities, particularly their findings on "Acute effects of whole-body vibration on running gait in marathon runners" which demonstrates how physical activity interventions affect neuromuscular function. This reference would enhance the mechanistic discussion in the introduction and provide context for the nonlinear muscle responses observed.

The manuscript would benefit significantly from incorporating Migliorini et al.'s systematic review on "Management of transient bone osteoporosis" from the British Medical Bulletin, which provides crucial insights into the relationship between physical inactivity and musculoskeletal health outcomes. This reference would strengthen the discussion of the Sarcopenic Index findings and provide clinical context for the observed threshold effects.

For the hemodynamic components of the study, the authors should integrate findings from Russo et al.'s research on "Day-time effect on postural stability in young sportsmen" published in MLTJ, which examines cardiovascular-musculoskeletal interactions. This would enhance the theoretical framework linking vascular and muscle phenotypes and provide mechanistic support for the coordinated "muscle-fat-vascular" axis response proposed by the authors.

The fat distribution findings would be substantially strengthened by incorporating insights from ÄŒular et al.'s work on "The prevalence of Constituent Year effect in Youth Olympic Games: implications for talent identification and development in basketball" from Acta Kinesiologica, which, while focused on athletic populations, provides important data on body composition patterns across different demographic groups. This reference would enhance the discussion of the demographic modifiers observed in the FDI associations.

Finally, to address the limitations of cross-sectional design and enhance the discussion of intervention implications, the authors should incorporate Padulo et al.'s longitudinal work on "Training During the COVID-19 Lockdown: Knowledge, Beliefs, and Practices of 12,526 Athletes from 142 Countries and Six Continents," which provides insights into sedentary behavior changes and their health implications over time. This would strengthen the concluding discussion about intervention strategies and future research directions.

Technical Quality Assessment

The figures demonstrate clear conceptual thinking but would benefit from enhanced graphical quality and more intuitive presentations of the complex spline relationships. Figure 1's theoretical framework is well-conceived but could be more precisely linked to the specific indices used. The restricted cubic spline plots in Figure 4 are informative but challenging to interpret without clearer confidence intervals and clinical reference points. Table presentations are comprehensive but could improve readability through strategic use of bold formatting for significant findings and clearer delineation of the hierarchical model adjustments.

The English language quality is generally strong but occasionally suffers from overly complex sentence structures that may obscure key findings. Terms like "multisystem phenotype" and "coordinated muscle-fat-vascular axis" require clearer operational definitions. The manuscript would benefit from more direct, declarative statements about the clinical implications of the identified thresholds.

Recommendations for Enhancement

The manuscript requires fundamental strengthening in several areas. Most critically, the authors must acknowledge more explicitly that their cross-sectional design precludes any claims about "chronic stress responses" or "pathophysiological trajectories." The discussion should focus more conservatively on associations rather than implied causation. The composite indices need better justification and comparison to established clinical measures. Sample size calculations should be provided retrospectively to establish the reliability of subgroup findings. The clinical interpretation of the 10.73-hour threshold requires more cautious presentation, acknowledging the uncertainty inherent in self-reported exposure data.

Complete List of Suggested Citations:

ÄŒular, D., Miletic, A., & Babic, M. (2024). The prevalence of Constituent Year effect in Youth Olympic Games: implications for talent identification and development in basketball. Acta Kinesiologica, 18(1), 4–8.

Author Response

(The authors gave the same response as above.)

Reviewer 3 Report

Comments and Suggestions for Authors

Dear Authors,

I have reviewed your manuscript and consider the topic relevant and of interest to Healthcare. The study is well-grounded, presents a clear prospective design, and makes appropriate use of data from four consecutive NHANES cycles. However, to strengthen its contribution and clarity, I suggest the following improvements:

Abstract: Reformat it following the IMRaD structure, incorporating numerical data, making it explicit that this is a prospective design, and specifying the population and main variables analyzed. In scientific articles, especially in journals indexed in Healthcare (MDPI), it is essential to include concrete figures to support statements, as this allows readers to quickly assess the magnitude and relevance of the findings.

Keywords: Avoid repeating terms already present in the title and ensure they correspond to standardized descriptors (MeSH and DeCS), which will enhance the article’s visibility and retrieval.

Results: Add an initial comparative table presenting, for each survey year, the values of each variable along with their respective p-values, effect sizes (ES), and confidence intervals (CI). This will make it easier to clearly visualize the differences and the magnitude of the effects across cycles.

Discussion: At the end, include a section explicitly addressing the limitations (e.g., use of secondary data, possible self-report biases), strengths (large population-based dataset, prospective design), practical implications for public health, and future research directions.

Conclusion: Currently, the conclusion is general and does not directly and explicitly address the objective stated in the introduction. I recommend reformulating it to synthesize the key findings in light of the study’s objective and initial hypothesis, avoiding extrapolations not supported by the results.

These adjustments will improve the methodological quality, internal coherence, and international impact of the manuscript.

Author Response

(The authors gave the same response as above.)
